# Aggregating explanation methods for neural networks stabilizes explanations

## Abstract

Despite a growing literature on explaining neural networks, no consensus has been reached on how to explain a neural network decision or how to evaluate an explanation. Our contributions in this paper are twofold. First, we investigate schemes to combine explanation methods and reduce model uncertainty to obtain a single aggregated explanation. The aggregation is more robust and aligns better with the neural network than any single explanation method. Secondly, we propose a new approach to evaluating explanation methods that circumvents the need for manual evaluation and is not reliant on the alignment of neural networks and humans decision processes.

## 1 Introduction

Despite the great success of neural networks especially in classic visual recognition problems, explaining the networks' decisions remains an open research problem Samek et al. (2019). This is due in part to the complexity of the visual recognition problem and in part to the basic 'ill-posedness' of the explanation task. This challenge is amplified by the fact that there is no agreement on what a sufficient explanation is and how to evaluate an explanation method.

Many different explanation strategies and methods have been proposed (Simonyan et al., 2013; Zeiler & Fergus, 2014; Bach et al., 2015; Selvaraju et al., 2017; Smilkov et al., 2017; Sundararajan et al., 2017). Focusing on visual explanations for individual decisions, most methods either use a backpropagation approach or aim to construct a simpler linear model with an intuitive explanation. The plethora of explanation approaches is a signature of the high-level epistemic uncertainty of the explanation task.

This paper is motivated by a key insight in machine learning: Ensemble models can reduce both bias and variance compared to applying a single model. A related approach was pursued for *functional* visualization in neuroimaging (Hansen et al., 2001). Here we for the first time explore the potential of aggregating explanations of individual visual decisions in reducing epistemic uncertainty for neural networks.

We test the hypothesis that ensembles of multiple explanation methods are more robust than any single method. This idea is analyzed theoretically and evaluated empirically. We discuss the properties of the aggregate explanations and provide visual evidence that they combine features, hence are more complete and less biased than individual schemes. Based on this insight, we propose two ways to aggregate explanation methods, *AGG-Mean* and *AGG-Var*. In experiments on Imagenet, MNIST, and FashionMNIST, the aggregates identify relevant parts of the image more accurately than any single method.

Second, we introduce IROF (**I**terative **R**emoval **O**f **F**eatures) as a new approach to quantitatively evaluate explanation methods without relying on human evaluation. We circumvent the problems of high correlation between neighbor pixels as well as the human bias that are present in current evaluation methods.

## 2 RELATED WORK

### 2.1 EXPLANATION METHODS

The open problem of explainability is reflected in a lot of recent work (Kindermans et al., 2017; Selvaraju et al., 2017; Bach et al., 2015; Zhang et al., 2018; Zhou et al., 2016; Ancona et al., 2018; Ribeiro et al., 2016; Rieger et al., 2018; Kim et al., 2018; Lundberg & Lee, 2017; Zintgraf et al., 2017; Simonyan et al., 2013; Zeiler & Fergus, 2014; Selvaraju et al., 2017; Smilkov et al., 2017; Sundararajan et al., 2017; Shrikumar et al., 2017; Montavon et al., 2017; Chang et al., 2018). We focus on generating visual explanations for single samples. The first work in this direction was Simonyan et al. (2013) with *Saliency Maps (SM)* that proposed backpropagating the output onto the input to gain an understanding of a neural network decision. The relevance for each input dimension is extracted by taking the gradient of the output w. r. t. to the input. This idea was extended by Springenberg et al. (2014) into *Guided Backpropagation (GM)* by applying ReLU non-linearities after each layer during the backpropagation. Compared to Saliency, this removes visual noise in the explanation. *Grad-CAM (GC)* from Selvaraju et al. (2017) is an explanation method, developed for use with convolutional neural networks. By backpropagating relevance through the dense layers and up-sampling the evidence for the convolutional part of the network, the method obtains coarse heatmaps that highlight relevant parts of the input image. *Integrated Gradients (IG)* Sundararajan et al. (2017) sums up the gradients from linearly interpolated pictures between a baseline, e.g. a black image, and the actual image. *SmoothGrad (SG)* filters out noise from a basic saliency map by creating many samples of the original input with Gaussian noise Smilkov et al. (2017). The final saliency map is the average over all samples.

Finally, we also consider *LIME* Ribeiro et al. (2016). In contrast to the other methods, *LIME* is not based on backpropagation. Instead, it approximates the neural network with a linear model locally around the input to be explained. The coefficients of the linear model for the respective input dimensions give the importance of each dimension. Compared to the other methods this is much more computationally expensive as it requires many passes through the neural network.

### 2.2 EVALUATION OF EXPLANATION METHODS

The evaluation of explanation methods is a relatively recent topic with few systematic approaches (Bau et al., 2017; Ancona et al., 2018; Hooker et al., 2018; Adebayo et al., 2018; Fong & Vedaldi, 2017). To our knowledge, Bach et al. (2015) proposed the first quantitative approach to evaluate an explanation method by flipping pixels to their opposite and comparing the decrease in output with the relevance attributed to the pixel by the explanation method. As the authors note, this only works for low-dimensional input. This work was followed up upon in Samek et al. (2016). By dividing high-dimensional images into squares, they make the method feasible for high-dimensional inputs. Squares with high relevance (as measured by the explanation method) consecutively get replaced with noise sampled from the uniform distribution. The difference between the original output and the output for the degraded images indicates the quality of the explanation method.

Hooker et al. (2018) proposes another quantitative approach to evaluate explanation methods called ROAR. For each explanation method they extract the relevance maps over the entire training set. They degrade the training set by setting different percentages of the pixels with the highest relevance to the mean and retrain the network. Each retrained network is evaluated on the test set. The accuracy on the test set decreases dependent on the percentage of pixels set to the mean. This requires retraining the same architecture multiple times for each explanation method at a high computational cost.

Ancona et al. (2018) proposed a different approach to evaluate explanation methods, called Sensitivity-$n$, based on the notion that the decrease in output when a number of inputs are canceled out should be equal to the sum of their relevances.
For a range of $n$ (between 1 and the total number of inputs) they sample a hundred subsets of the input. For each $n$, the Pearson Correlation Coefficient (PCC) between the decrease in output, when the subset of features is removed, and the sum of their relevances is reported. The result is a curve of the PCC dependent on the percentage of the input being removed. For a good explanation method, the PCC will decrease slowly.

## 3 METHODS

### 3.1 AGGREGATING EXPLANATION METHODS TO REDUCE VARIANCE

All currently available explanation methods have weaknesses that are inherent to the approach and include significant uncertainty in the resulting heatmap (Kindermans et al., 2017; Adebayo et al., 2018; Smilkov et al., 2017). A natural way to mitigate this issue and reduce noise is to combine multiple explanation methods. Ensemble methods have been used for a long time to reduce the variance and bias of machine learning models. We apply the same idea to explanation methods and build an ensemble of explanation methods.

We assume a neural network $F : X \mapsto y$ with $X \in \mathbb{R}^{m \times m}$ and a set of explanation methods $\{e_j\}_{j=1}^{J}$ with $e_j : X, y, F \mapsto E$ with $E \in \mathbb{R}^{m \times m}$. We write $E_{j,n}$ for the explanation obtained for $X_n$ with method $e_j$ and denote the mean aggregate explanation as $\bar{e}$ with $\bar{E}_n = \frac{1}{J} \sum_{j=1}^{J} E_{j,n}$. While we assume the input to be an image $\in \mathbb{R}^{m \times m}$, this method is generalizable to inputs of other dimensionalities as well.

To get a theoretical understanding of the benefit of aggregation, we hypothesize the existence of a 'true' explanation $\hat{E}_n$. This allows us to quantify the error of an explanation method as the mean squared difference between the 'true' explanation and an explanation procured by an explanation method, i.e. the MSE.

For clarity we subsequently omit the notation for the neural network. We write the error of explanation method $j$ on image $X_n$ as $\mathrm{err}_{j,n} = ||E_{j,n} - \hat{E}_n||^2$ with

$$\mathrm{MSE}(E_j) = \frac{1}{N} \sum_n \mathrm{err}_{j,n}$$

and $\mathrm{MSE}(\bar{E}) = \frac{1}{N} \sum_n ||\bar{E}_n - \hat{E}_n||^2$ is the MSE of the aggregate. The typical error of an explanation method is the mean error over all explanation methods

$$\overline{\mathrm{MSE}} = \frac{1}{J} \sum_j \mathrm{MSE}(E_j).$$

With these definitions we can do a standard bias-variance decomposition (Geman et al., 1992). Accordingly we can show the error of the aggregate will be less that the typical error of explanation methods,

$$\overline{\mathrm{MSE}} \qquad = \frac{1}{N} \sum_n \frac{1}{J} \sum_j ||\hat{E}_n - E_{j,n}||^2 \qquad (1)$$
$$= \frac{1}{N} \sum_n ||\hat{E}_n - \bar{E}_n||^2 + \frac{1}{NJ} \sum_{n,j} ||\bar{E}_n - E_{j,n}||^2,$$

hence,

$$\overline{\mathrm{MSE}} = \frac{1}{J} \sum_j \frac{1}{N} \sum_n \underbrace{||\bar{E}_n - E_{j,n}||^2}_{\text{epistemic uncertainty}} + \mathrm{MSE}(\bar{E}) \geq \mathrm{MSE}(\bar{E}).$$

A detailed calculation is given in appendix A.1. The error of the aggregate $\mathrm{MSE}(\bar{E})$ is less than the typical error of the participating methods. The difference - a 'variance' term - represents the epistemic uncertainty and only vanishes if all methods produce identical maps. By taking the average over all available explanation methods, we reduce the variance of the explanation compared to using a single method. To obtain this average, we normalize all input heatmaps such that the relevance over all pixels sum up to one. This reflects our initial assumption that all individual explanation methods are equally good estimators. We refer to this approach as *AGG-Mean*.

$$E_{\text{Agg-Mean},n} = \frac{1}{J} \sum_{j=1}^{J} E_{j,n}$$

This estimator however does not take into account the estimate of the local epistemic uncertainty, i.e. the disagreement between methods. A way to incorporate this information is to form an 'effect

size' map by dividing the mean aggregate locally with its standard deviation (Sigurdsson et al., 2004). Intuitively, this will assign less relevance to segments with high disagreement between methods.

For stability, we divide not directly by the local variance but add a constant $\epsilon$ to the estimate of the local variance. This can be interpreted as a smoothing regularizer or a priori information regarding epistemic and aleatoric uncertainties. We refer to this approach as *AGG-Var*.

$$E_{\text{AGG-Var},n} = \frac{1}{J} \sum_{j=1}^{J} \frac{E_{j,n}}{\sigma(E_{j \in J,n}) + \epsilon}$$

where $\sigma(E_{j \in J,n})$ is the point-wise standard deviation over all explanations $j \in J$ for $X_n$

In section 4 we will evaluate and compare *AGG-Mean* and *AGG-Var* against basic explanation methods.

## 3.2 EVALUATING EXPLANATION METHODS QUANTITATIVELY WITH IROF

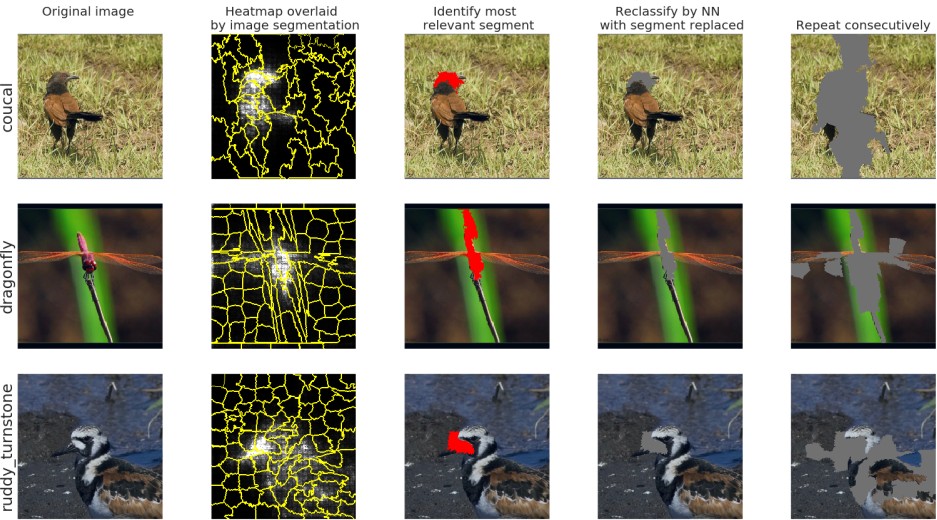

Figure 1: Quantitative evaluation with IROF: Relevant segments as identified by an explanation method get consecutively replaced by the mean colour over the entire dataset. The IROF score of an explanation method is the integrated decrease in the class score over the number of removed segments.

Quantitative evaluation is a recurring problem with explainability methods. This is especially true for high-dimensional input, such as images, where important features are made up of locally highly correlated pixels. If the information in one pixel is lost, this will not change the overall feature and should therefore not result in a changed output score. The relevance values of single pixels are not indicative of the feature's importance as a whole. We circumvent this problem by utilizing conventional image segmentation, a well-explored area. By first dividing the image into coherent segments, we avoid the interdependency between the inputs.

**Methodology** We assume a neural network $F : X \mapsto y$ with $X \in \mathbb{R}^{m \times m}$ and a set of explanation methods $\{e_j\}_{j=1}^{J}$ with $e_j : X, y, F \mapsto E$ with $E \in \mathbb{R}^{m \times m}$.

Furthermore we partition each image $X_n$ into a set of segments $\{S_n^l\}_{l=1}^{L}$ using a given segmentation method with $s_{n,i,j}^l = 1$ indicating that pixel $x_{n,i,j}$ belongs to segment $l$. Computing the mean importance $\frac{||E_{j,n} S_n^l||_1}{||S_n^l||_1}$ of each segment according to a given explanation method $j$, two segments can be directly compared against each other. By sorting the segments in decreasing order of importance according to the explanation method, we get a ranking of how relevant each segment of the image is.

We use $X_n^{'l}$ to indicate $X_n$ with the $l$ segments with highest mean relevance replaced with the mean value. Taking $F(X_n^{'l})_y$ repeatedly with increasing $l \in 0, ..., L$ results in a curve of the class score dependent on how many segments of the image are removed. Dividing this curve by $F(X_n^{'0})_y$ normalizes the scores to be within $[0, 1]$ and makes curves comparable between input samples and networks.

If an explanation method works well, it will attribute high relevance to segments important for classification. As segments with high relevance are removed first, the score for the target class will go down faster. By computing the area over the curve (AOC) for the class score curve and averaging over a number of input samples, we can identify the method that identifies relevant areas of the image more reliably. For a good explanation method, the AOC will be higher. We refer to this evaluation method as the **iterative removal of features (IROF)**. The IROF for a given explanation method $e_j$ is expressed as:

$$\text{IROF}(e_j) = \sum_{n=1}^{N} \text{AOC} \left( \frac{F(X_n^{'l})_y}{F(X_n^{'0})_y} \right)_{l=0}^{L}$$

This approach is a quantitative comparison of two or more explainability methods that does not rely on human evaluation or alignment between human and neural network reasoning. For each explanation method the workflow produces a single value, enabling convenient comparison between two or more explanation methods. If the AOC is higher, the explanation method captures more information about the neural network classification.

IROF is dependent on having meaningful segments in the input, as natural images do. Dividing up text or non-natural images such as EEG into meaningful and independent segments does not have a natural solution and is left for future research.

## 4 EXPERIMENTS

We first present empirical validation of our proposed evaluation technique IROF in section 4.2. Subsequently we evaluate the aggregation of explanation techniques against the vanilla techniques with IROF, Sensitivity-$n$ and qualitative evaluation. In appendix A.6.1 we compare aggregated methods on a dataset of human-annotated heatmaps.

### 4.1 EXPERIMENTAL DETAILS

We tested our method on five neural network architectures that were pre-trained on ImageNet: VGG19, Xception, Inception, ResNet50 and ResNet101 (Deng et al., 2009; Simonyan & Zisserman, 2014; He et al., 2016; Chollet, 2017; Szegedy et al., 2016). [1] Additionally, we ran experiments on CNN trained on the MNIST and FashionMNIST dataset LeCun & Cortes (2010); Xiao et al. (2017).

We compared the aggregation methods against Saliency (SM), Guided Backpropagation (GB), SmoothGrad (SG), Grad-CAM (GC) and Integrated Gradients (IG) to have a selection of attribution-based methods. Additionally we compared against LIME as a method that is not based on attribution but rather on local approximation Ribeiro et al. (2016). The aggregations are based on all attribution-based methods.

Some of the methods result in positive and negative evidence. We only considered positive evidence for the ImageNet tasks to compare methods against each other. To check that this does not corrupt the methods, we compared the methods that do contain negative results against their filtered version and found negligible difference between the two versions of a method in the used metrics.

For *Agg-Mean* we introduced an additional parameter, $\epsilon$ to the divisor. In our experiments we set $\epsilon$ to be ten times the mean $\sigma$ over the entire dataset.

---

[1]Models retrieved from https://github.com/keras-team/keras.

## 4.2 EVALUATING IROF FOR VALIDITY AS AN EVALUATION METHOD

A good evaluation method should be able to reject the null hypothesis (a given explanation method is no better than random choice) with high confidence. We use this as motivation to evaluate and compare IROF by calculating the paired t-test of an explanation method versus random guessing. This is done with multiple explanation methods and networks, to reduce the impact of the explanation method.

We compare IROF and pixel removal with mean value and black as a replacement value respectively. Additionally we compare against Samek et al. (2016) as explained in section 2. For IROF and Samek et al. (2016) we set the 10% most relevant segments to the mean value over the dataset. For pixel removal, we set the equivalent number of pixels to the mean value. The percentage of segments or pixels being removed was chosen ad hoc. If the difference in degradation between random choice and the explanation method is high, the explanation method reports meaningful information. Since we compare the same explanation method on the same neural network with different evaluation methods, the p-values only contain information about how meaningful the evaluation method is.

We computed IROF and pixel removal with black or mean replacement values and compared the p-value changes dependent on the number of samples. Results are shown in fig. 2 (extended in appendix A.6). In table 1 we provide results for forty images in tabular form for an easier overview (other methods in appendix A.6). On forty images, all evaluation methods produce p-values below 0.05. Thus, all evaluation methods can distinguish between random guessing and an explanation method.

However, IROF can reject the null hypothesis (the explanation method does not contain any information), with much higher confidence with the same number of samples for any configuration. We can conclude that IROF is more sensitive to the explanation method than pixel removal or Samek et al. (2016), making it the better choice to quantitatively evaluate an explanation method.

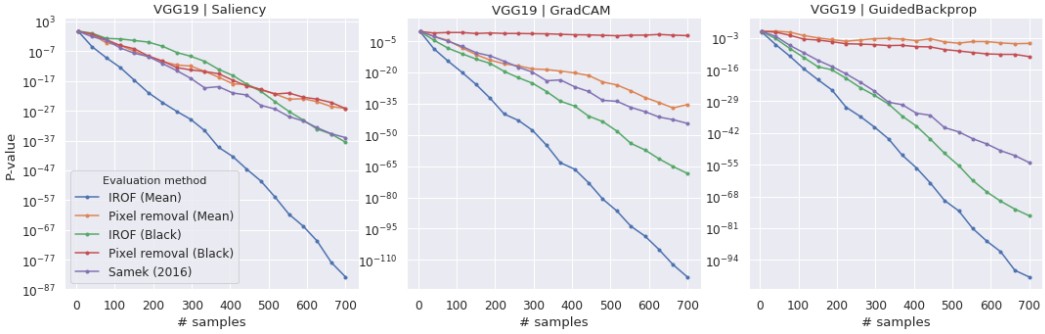

Figure 2: P-values for the rejection of the random removal null-hypothesis. Lower is better. P-values are on a logarithmic scale. IROF performs best in all scenarios.

Table 1: t-test: p-values of Random choice vs Saliency Mapping on forty images. All p-values < 0.05.

| EVALUATION METHOD | T-STAT | P-VAL |
|---|---|---|
| IROF (BLACK) | 1.58 | 1.22E-01 |
| IROF (MEAN) | 5.44 | 3.60E-06 |
| PIXEL FLIPPING (BLACK) | 1.92 | 6.22E-02 |
| PIXEL FLIPPING (MEAN) | 2.10 | 4.25E-02 |
| SAMEK (2016) | 2.77 | 8.65E-03 |

## 4.3 EVALUATING EXPLANATION METHODS WITH IROF

To quantitatively compare the quality of the explanation methods on a more challenging dataset, we use IROF on a number of neural network architectures trained on ImageNet. In table 2 we report

Table 2: IROF scores across methods and architectures. *AGG-Mean* and *AGG-Var* surpass all methods in all scenarios. All std $< 0.05$.

| METHOD | INCEPTION | RESNET101 | RESNET50 | VGG19 | XCEPTION |
|---|---|---|---|---|---|
| RANDOM | 60.3 | 61.0 | 65.5 | 68.4 | 50.1 |
| SOBEL | 71.5 | 74.5 | 75.6 | 79.6 | 68.3 |
| LIME | 74.0 | 67.6 | 66.7 | 73.4 | 74.0 |
| SM | 76.2 | 75.6 | 76.2 | 81.4 | 72.6 |
| GB | 73.7 | 77.6 | 80.9 | 84.3 | 74.8 |
| IG | 76.0 | 76.0 | 76.2 | 79.8 | 73.3 |
| SG | 77.9 | 77.7 | 77.5 | 83.9 | 75.0 |
| GC | 78.7 | 77.8 | 78.5 | 86.1 | 73.5 |
| AGG-MEAN | 79.9 | 81.1 | 80.9 | 86.6 | **76.7** |
| AGG-VAR | **80.4** | **81.2** | **81.0** | **86.7** | **76.7** |

the IROF as described in section 3.2. We include two non-informative baselines. *Random* randomly chooses segments to remove. *Sobel* is a sobel edge detector. Neither of them contain information about the neural network.

All explanation methods have a lower IROF than the random baseline on all architectures tested, indicating that all methods contain information about the image classification. Except for LIME, all methods also surpass the stronger baseline, *SOBEL*. The ranking of unaggregated methods varies considerably between architectures. This variance indicates that the accuracy of an explanation method depends on the complexity and structure of the neural network architecture. For all architectures *AGG-Mean* and *AGG-Var* have a lower IROF score than any non-aggregated method. For ResNet101 the difference between the best unaggregated method and the aggregated methods is especially large. We hypothesize, that the benefit of aggregating explanation methods increases for more complex neural network with large epistemic uncertainty on the explanation.

We can empirically confirm that aggregating methods improves over unaggregated methods and more reliably identifies parts of the images that are relevant for classification.

## 4.4 QUALITATIVE VISUAL EVALUATION

We show heatmaps for individual examination for each of the methods in fig. 3 and compare qualitatively (large version of fig. 3 in appendix A.6.2). While visual evaluation of explanations for neural networks can be misleading, there is no better way available of checking whether any given explanation method agrees with intuitive human understanding (Adebayo et al., 2018). Additionally, we compute alignment between human-annotated images and the explanation methods in **??**, using the human benchmark for evaluation introduced in Mohseni & Ragan (2018).

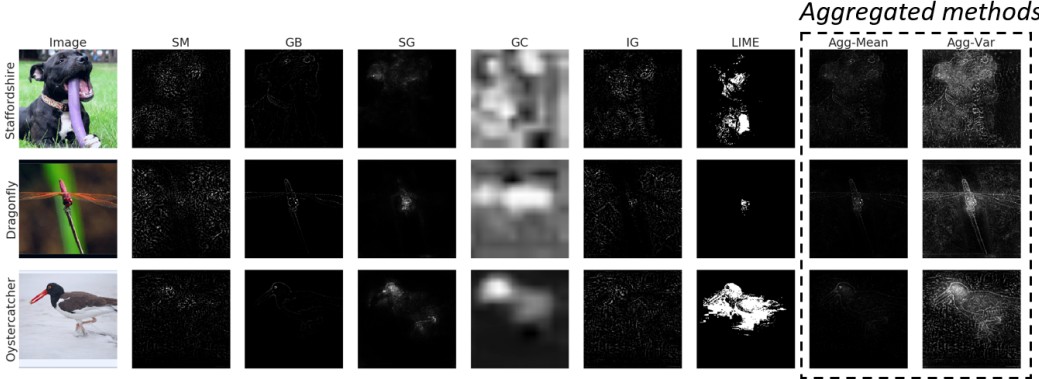

Figure 3: Example images from Imagenet and the heatmaps produced by different methods on VGG19. Aggregated methods combine features from all methods. Too heavy focus on one feature by SmoothGrad is smoothed away by the aggregation.

*AGG-Var* combines features of the aggregrated methods by attributing relevance on the classified object as a whole, but considering smaller details such as the face of an animal as more relevant. It is a combination of the detail-oriented and shape-oriented methods. Compared to SmoothGrad, which concentrates on one isolated feature, the relevance is more evenly distributed and aligned with our human intuition that classification takes context into account and does not rely on e.g. only the beak of a bird. We can conclude that combining explainability methods provides a meaningful visual improvement over single methods.

### 4.5 EVALUATION WITH SENSITIVITY-*n* ON LOW-DIMENSIONAL INPUT

To quantitatively compare explanation methods on a low-dimensional input we use Sensitivity-*n* (Ancona et al., 2018). The exact procedure is described in section 2. We compare on MNIST (LeCun & Cortes, 2010) and FashionMNIST (Xiao et al., 2017), two low-dimensional dataset with a basic CNN[2] (architecture in appendix) . We follow the procedure suggested in Ancona et al. (2018) and test on a hundred randomly sampled subsets for 1000 randomly sampled test images. The number of pixels in the set *n* is chosen at fifteen points logarithmically spaced between 10 and 780 pixels.

As described in section 2, for a range of $n$ (between 1 and the total number of inputs) a hundred subsets of the input features are removed. For each $n$, the average Pearson Correlation Coefficient (PCC) between the decrease in output and the relevance of the removed output features is reported. The result is a curve of the PCC dependent on the removed percentage.

We show results in fig. 4. *AGG-Mean* and *AGG-Var* perform in range of the best methods. For the CNN trained on FashionMNIST, *AGG-Mean* and *AGG-Var* perform better than unaggregated methods. For the CNN trained on MNIST, Guided Backpropagation and *AGG-Mean* perform best. For both networks (trained on FashionMNIST and MNIST respectively), SmoothGrad and GradCAM perform considerably worse than the other methods.

In summary, aggregation seems to not be as beneficial when applied to a low-dimensional, "easier" tasks such as MNIST as it is for ImageNet, performing in range of the best unaggregated method. We hypothesize that this is because there is less epistemic uncertainty in explanations for less complex tasks and network architectures.

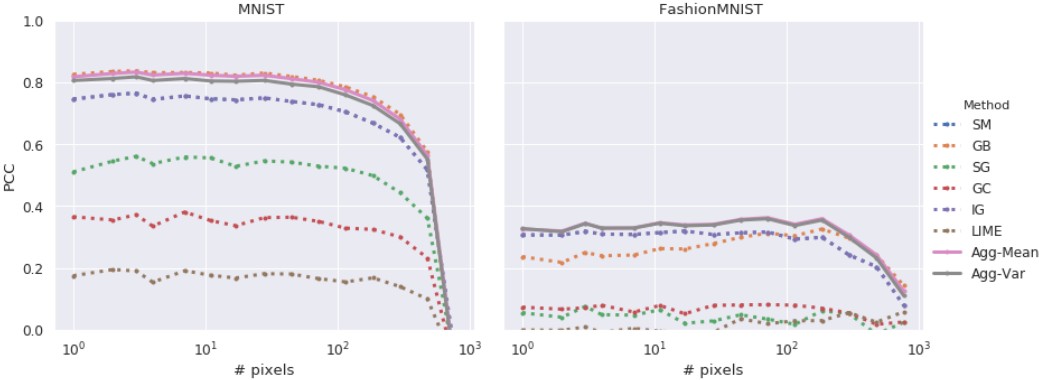

Figure 4: Sensitivity-*n* for explanation methods. Higher is better. The proposed methods, *AGG-Mean* and *AGG-Var* perform better or equally good as all other methods.

## 5 CONCLUSION

In this work we gave a simple proof that aggregating explanation methods will perform at least as good as the typical individual method. In practice, we found evidence that aggregating methods outperforms any single method. We found this evidence substantiated across quantitative metrics. While our results show that different vanilla explanation methods perform best on different network architectures, an aggregation supersedes all of them on any given architecture.

---

[2]Model and code retrieved from https://github.com/keras-team/keras/blob/master/examples/mnist_cnn.py.

Additionally we proposed a novel way of evaluation for explanation methods that circumvents the problem of high correlation between pixels and does not rely on visual inspection by humans, an inherently misleading metric.

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

# A   APPENDIX

## A.1   AGGREGATING EXPLANATION METHODS TO REDUCE VARIANCE - DETAILED DERIVATION

All currently available explanation methods have weaknesses that are inherent to the approach and include significant noise in the heatmap (Kindermans et al., 2017; Adebayo et al., 2018; Smilkov et al., 2017). A natural way to mitigate this issue and reduce noise is to combine multiple explanation methods. Ensemble methods have been used for a long time to reduce the variance and bias of machine learning models. We apply the same idea to explanation methods and build an ensemble of explanation methods.

We assume a neural network $F : X \mapsto y$ with $X \in \mathbb{R}^{m x m}$ and a set of explanation methods $\{e_j\}_{j=1}^{J}$ with $e_j : X, y, F \mapsto E$ with $E \in \mathbb{R}^{m x m}$. We write $E_{j,n}$ for the explanation obtained for $X_n$ with method $e_j$ and denote the mean aggregate explanation as $\bar{e}$ with $\bar{E}_n = \frac{1}{J} \sum_{j=1}^{J} E_{j,n}$. While we assume the input to be an image $\in R^{m x m}$, this method is generalizable to inputs of other dimensions as well.

We define the error of an explanation method as the mean squared difference between a hypothetical 'true' explanation and an explanation procured by the explanation method, i.e. the MSE. For this definition we assume the existence of the hypothetical 'true' explanation $\hat{E}_n$ for image $X_n$.

For clarity we subsequently omit the notation for the neural network.

We write the error of explanation method $j$ on image $X_n$ as $err_{j,n} = ||E_{j,n} - \hat{E}_n||^2$ with

$$\text{MSE}(E_j) = \frac{1}{N} \sum_n err_{j,n}$$

and $\text{MSE}(\bar{E}) = \frac{1}{N} \sum_n ||\bar{E}_n - \hat{E}_n||^2$ is the MSE of the aggregate. The typical error of an explanation method is represented by the mean

$$\overline{\text{MSE}} = \frac{1}{N} \sum_n \frac{1}{J} \sum_j ||\hat{E}_n - E_{j,n}||^2$$

$$= \frac{1}{NJ} \sum_{n,j} ||\hat{E}_n - E_{j,n} + \bar{E}_n - \bar{E}_n||^2$$

$$= \frac{1}{NJ} \sum_{n,j} ||(\hat{E}_n - \bar{E}_n) + (\bar{E}_n - E_{j,n})||^2$$

$$= \frac{1}{NJ} \sum_{n,j} \left( ||\hat{E}_n - \bar{E}_n||^2 + ||\bar{E}_n - E_{j,n}||^2 + 2\text{Tr}\left[ (\hat{E}_n - \bar{E}_n)(\bar{E}_n - E_{j,n}) \right] \right)$$

$$= \frac{1}{N} \sum_n ||\hat{E}_n - \bar{E}_n||^2 + \frac{1}{NJ} \sum_{n,j} ||\bar{E}_n - E_{j,n}||^2 + 2\frac{1}{N} \sum_n \text{Tr}\left[ (\hat{E}_n - \bar{E}_n) \left( \frac{1}{J} \sum_j (\bar{E}_n - E_{j,n}) \right) \right]$$

$$= \frac{1}{N} \sum_n ||\hat{E}_n - \bar{E}_n||^2 + \frac{1}{NJ} \sum_{n,j} ||\bar{E}_n - E_{j,n}||^2 + 2\frac{1}{N} \sum_n \text{Tr}\left[ (\hat{E}_n - \bar{E}_n) \underbrace{\frac{1}{J} \sum_j (\bar{E}_n - E_{j,n})}_{=0} \right]$$

$$= \frac{1}{N} \sum_n ||\hat{E}_n - \bar{E}_n||^2 + \frac{1}{NJ} \sum_{n,j} ||\bar{E}_n - E_{j,n}||^2,$$

hence,

$$\overline{\text{MSE}} = \text{MSE}(\bar{E}) + \frac{1}{NJ} \sum_{n,j} \underbrace{||\bar{E}_n - E_{j,n}||^2}_{\text{epistemic uncertainty}} \geq \text{MSE}(\bar{E})$$

The error of the aggregate $\text{MSE}(\bar{E})$ is less than the typical error of the participating methods. The difference - a 'variance' term - represents the epistemic uncertainty and only vanishes if all methods produce identical maps.

## A.2 Comparing aggregate of two methods

In section 3.1 we showed theoretically that the average MSE of two or more explanation methods will always be higher than the error of the averaged of those methods. Empirically, we test this for IROF with combinations of any two methods for ResNet101 and show the results in fig. 5. For any two methods, the matrix shows the ratio between the aggregate method IROF and the average IROF of the aggregated methods. The aggregate IROF is always lower, confirming our theoretical results.

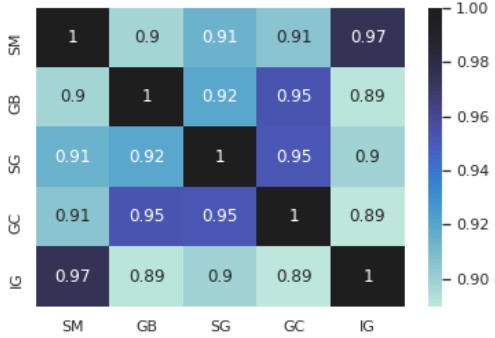

Figure 5: Ratios between the aggregate IROF and the averaged IROF of two methods. Aggregation always improves on the results, as all values outside of the diagonal are below one.

## A.3 IROF extended

## A.4 Choosing explanation methods

An important part of our method is the choice of explanation methods to be included in the aggregation. In our work we focus on backpropagation-based methods, since they tend to be computationally cheap. In our opinion this makes for a more realistic use case, not only for human but also for machine post processing.

In contrast, locality based methods such as LIME or the method by Fong & Vedaldi (2017) require many forward passes, since their methods essentially "learn" what parts of the input are relevant. We included LIME in our experiments to have a not-backpropagation based method included.

## A.5 Experimental setup

### A.5.1 General

We use SLIC for image segmentation due to availability and quick run time (Achanta et al., 2012). Preliminary experiments with Quickshift showed similar results (Vedaldi & Soatto, 2008). SLIC was chosen over Quickshift due to the quicker run time. The number of segments was set to 300 ad hoc. fig. 1 shows the same procedure with 100 segments for the sake of clarity.
For AGG-Var, we add a constant to the denominator. We set this constant to 10 times the mean std, a value chosen empirically after trying values in the range of $[1, 10, 100]$ times the mean.
Evaluations were run with a set random seed for reproducibility. Stddev were reported either for each individual result or if they were non-significant in the caption to avoid cluttering the results.
Since our work does not include computationally heavy training, we did not record the exact computing infrastructure.

### A.5.2 MNISTs

The training for both models was equivalent. The architecture was as follows:
(input)-(conv(32,3,3))-(conv(64,3,3))-(maxPool(2,2))-(dropout(0.25))
-(fully connected(128))-(dropout(0.5))-(output(10))

ReLU was used as a non-linearity for both. All networks were trained with Adadelta and early stopping on the validation set (patience of three epochs) Zeiler (2012). The final accuracy for MNIST was 99.21%.: The final accuracy on FashionMNIST was 92.46%.

### A.5.3 IMAGENET

We tested our method on five network architectures that were pre-trained on ImageNet: VGG19, Xception, Inception, ResNet50 and ResNet101 (Deng et al., 2009; Simonyan & Zisserman, 2014; He et al., 2016; Chollet, 2017; Szegedy et al., 2016). We used the pre-trained networks VGG19, Xception and Inception, obtained from the keras library and did not change the networks in any way. (Deng et al., 2009; Szegedy et al., 2016; Chollet, 2017; Simonyan & Zisserman, 2014).

We downloaded the data from the ImageNet Large Scale Visual Recognition Challenge website and used the validation set only. No images were excluded. The images were preprocessed to be within $[-1, 1]$ unless a custom range was used for training (indicated by the preprocess function of keras).

## A.6 EVALUATING THE EVALUATION

We report p-values for evaluating with 50 images on ResNet101 in the manner described in section 4.2 in tabular form to provide a clear overview.

Table 3: t-test p-values of explanation methods for SmoothGrad.

| EVALUATION METHOD | T STATISTIC | P-VALUE |
|---|---|---|
| IROF (BLACK) | 3.97 | 3.22E-04 |
| IROF (MEAN) | 5.99 | 6.42E-07 |
| PIXEL FLIPPING (BLACK) | 0.55 | 5.86E-01 |
| PIXEL FLIPPING (MEAN) | 5.00 | 1.39E-05 |
| SAMEK (2016) | 2.97 | 5.17E-03 |

Table 4: t-test p-values of explanation methods for Guided backpropagation.

| EVALUATION METHOD | T STATISTIC | P-VALUE |
|---|---|---|
| IROF (BLACK) | 3.79 | 5.42E-04 |
| IROF (MEAN) | 5.64 | 1.93E-06 |
| PIXEL FLIPPING (BLACK) | 1.16 | 2.52E-01 |
| PIXEL FLIPPING (MEAN) | 0.49 | 6.30E-01 |
| SAMEK (2016) | 3.02 | 4.54E-03 |

Table 5: t-test p-values of explanation methods for Saliency.

| EVALUATION METHOD | T-STAT | P-VAL |
|---|---|---|
| IROF (BLACK) | 1.58 | 1.22E-01 |
| IROF (MEAN) | 5.44 | 3.60E-06 |
| PIXEL FLIPPING (BLACK) | 1.92 | 6.22E-02 |
| PIXEL FLIPPING (MEAN) | 2.10 | 4.25E-02 |
| SAMEK (2016) | 2.77 | 8.65E-03 |

Table 6: t-test p-values of explanation methods for GradCAM.

| EVALUATION METHOD | T STATISTIC | P-VALUE |
|---|---|---|
| IROF (BLACK) | 4.73 | 3.23E-05 |
| IROF (MEAN) | 7.81 | 2.42E-09 |
| PIXEL FLIPPING (BLACK) | -1.75 | 8.85E-02 |
| PIXEL FLIPPING (MEAN) | 3.26 | 2.38E-03 |
| SAMEK (2016) | 3.32 | 2.04E-03 |

We provide an extended version of fig. 2.

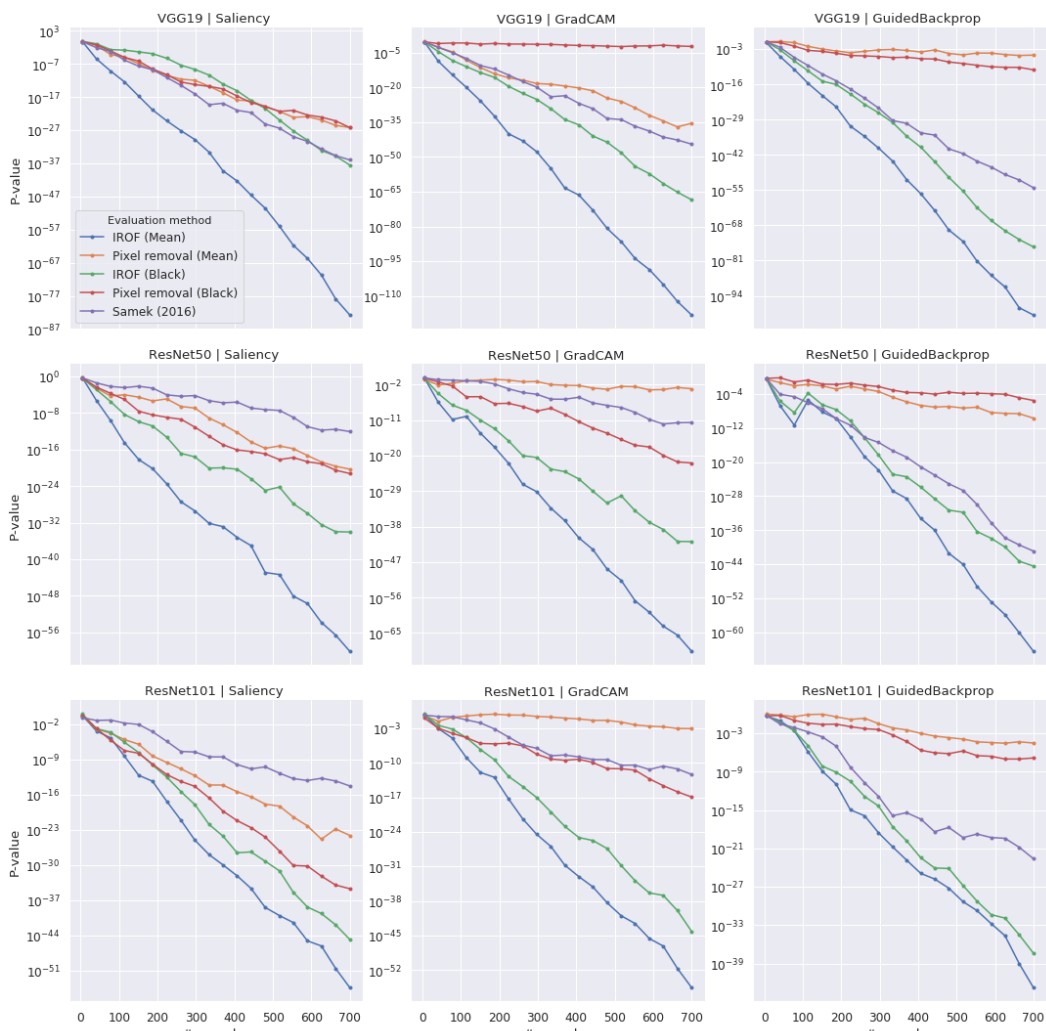

Figure 6: P-values for the rejection of the random removal null-hypothesis. Extended graphs

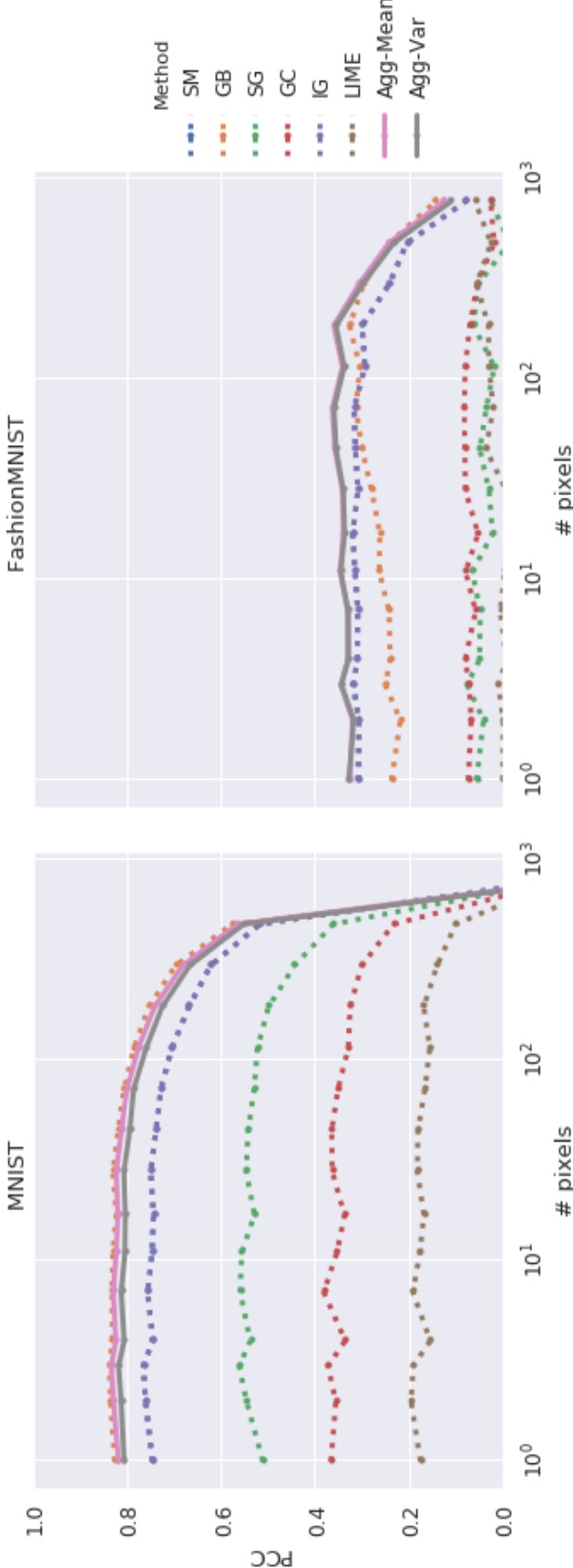

Figure 7: Sensitivity-*n* for explanation methods. Higher is better. The proposed methods, *AGG-Mean* and *AGG-Var* perform better or equally good as all other methods. (large version of fig. 4)

### A.6.1 ALIGNMENT BETWEEN HUMAN ATTRIBUTION AND EXPLANATION METHODS

We want to quantify whether an explanation method agrees with human judgement on which parts of an image should be important. While human annotation is expensive, there exists a benchmark for human evaluation introduced in Mohseni & Ragan (2018). The benchmark includes ninety images of

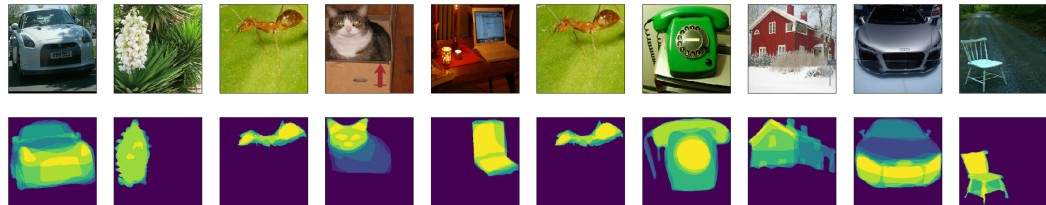

Figure 8: Example images from Mohseni & Ragan (2018) with human-annotated overlays.

categories in the ImageNet Challenge (ten images were excluded due to the category not being in the ImageNet challenge) and provides annotations of relevant segments that ten human test objects found important. Example images are shown in fig. 8.

While human evaluation is not a precise measure, we still expect some correlation between neural network and human judgement.

To test the alignment, we calculate the cosine similarity,

$$\text{similarity}(e_j) = \frac{\sum_{n=1}^{N} A_n E_{j,n}}{\sqrt{\sum_{n=1}^{N} A_n^2}\sqrt{\sum_{n=1}^{N} E_{j,n}^2}}$$

between the human annotation and the explanations produced by the respective explanation methods. $A_n$ is the human annotation of what is important for image $X_n$

Since the images in this dataset are 224x224 pixel large, we only compute the cosine similarity for the network architectures where pretrained networks with this input size were available.

We see that *AGG-Mean* and *AGG-Var* perform on-par with the best methods (SmoothGrad and GradCAM). While the aggregated methods perform better than the average explanation method, they do not surpass the best method.

When we combine the two best-performing single methods, SmoothGrad and GradCAM, we surpass each individual method. We hypothesize that this is because the epistemic uncertainty is reduced by the aggregate.

Table 7: Cosine similarity between heatmap and human annotated benchmark. All std below 0.05

| METHOD | RESNET101 | RESNET50 | VGG19 |
|---|---|---|---|
| AGG-MEAN | 0.63 | 0.66 | 0.64 |
| AGG-VAR | 0.66 | 0.68 | 0.67 |
| GB | 0.42 | 0.49 | 0.47 |
| GC | 0.60 | 0.62 | 0.60 |
| IG | 0.45 | 0.45 | 0.47 |
| MEAN(SG+GC) | **0.69** | **0.70** | **0.65** |
| SG | 0.63 | 0.65 | 0.59 |
| SM | 0.45 | 0.45 | 0.47 |

A.6.2 EXAMPLE HEATMAPS

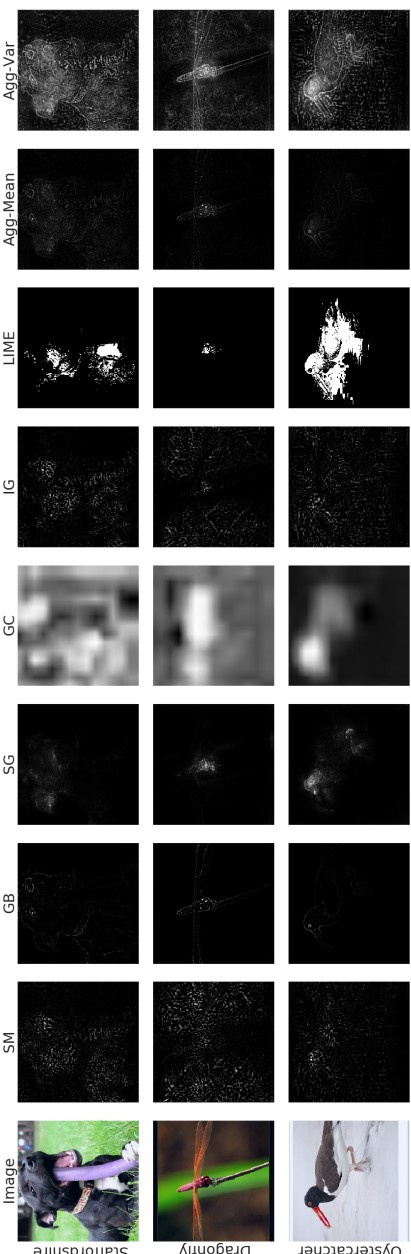

Figure 9: Example images from Imagenet and the heatmaps produced by different methods on VGG19. Aggregated methods combine features from all methods. Too heavy focus on one feature by SmoothGrad (f.e. the beak in lower row) is smoothed away by the aggregation.

