# OpenReview forum: "Aggregating explanation methods for neural networks stabilizes explanations"
_ICLR.cc/2020/Conference — Reject_

### Official Review · AnonReviewer1 · 2019-10-21
**Official Blind Review #1**

**Rating:** 8

**Review:**

The paper presents a study on explanation methods, proposing an interesting way to aggregate their results and providing empirical evidence that aggregation can improve the quality of the explanations.

The paper considered only methods using CNN for classifying images, leaving other applications for future investigation.

The results show aggregation of different explanation methods leads to better explanations, and is therefore exploitable, for high dimensional images while degrades with low-dimensional ones. In the latter case, it happens that the aggregated explanation explains a bit worse than the best non-aggregated explanation (from the graph it seems a very small difference though). This is odd because I would have assumed to see an improvement (at least a small one) using information from more than one system.

The paper also presents a score for evaluating explanation methods, which shows good results.

The paper is interesting and well written, the experimental campaign extensive enough and the methods are presented clearly.

There are some minor problems to be solved:
- In the abstract, there are two periods before the last sentence.
- When defining the size of matrices I suggest using \times instead of x to improve readability
- The caption in Figure 1 says "The decrease of the class score over the number of removed segments is reported.", but I cannot find where this decrease is reported.
- On page 5, in the parentheses it is reported that a given explanation method is no better than random choice. I would say that it is better (or not worse) than random choice, otherwise the methods would not provide any useful information.
- On page 6, forty images are considered in the text while fifty images are considered in the caption of Table 1. Please align the number with the correct one.

**Experience Assessment:**

I have read many papers in this area.

**Review Assessment: Checking Correctness Of Derivations And Theory:**

N/A

**Review Assessment: Checking Correctness Of Experiments:**

I carefully checked the experiments.

**Review Assessment: Thoroughness In Paper Reading:**

I read the paper at least twice and used my best judgement in assessing the paper.

---

> ### Author Response · Authors · 2019-11-09
> **Author's response**
>
> We would like to thank the reviewer for their thoughtful comments. We address their comments below.
> -          “In the latter case, it happens that the aggregated explanation explains a bit worse than the best non-aggregated explanation (from the graph it seems a very small difference though). This is odd because I would have assumed to see an improvement (at least a small one) using information from more than one system.”
> In fig. 4 we show that Agg-Mean and Agg-Var  slightly outperform all vanilla methods for FashionMNIST, which is equally low-dimensional as MNIST. In contrast to FashionMNIST, MNIST is unique due to the binary data distribution (black and white). By nature of this binary distribution, removing a pixel (e.g., setting it to black) is "informative". We hypothesize that this forced choice makes Sensitivity-n, proposed by [1], less reliable as an evaluation method for MNIST. This is not likely to be an issue in most real world data sets .
> -          “ The caption in Figure 1 says "The decrease of the class score over the number of removed segments is reported.", but I cannot find where this decrease is reported.”
> Thank you!  The IROF score is the integration of the class score over the number of segments. Based on your suggestion we changed the caption to “The IROF score of an explanation method is the integrated decrease in the class score over the number of removed segments.”
> -          “- On page 5, in the parentheses it is reported that a given explanation method is no better than random choice. I would say that it is better (or not worse) than random choice, otherwise the methods would not provide any useful information.”
> The quote in question: “A good evaluation method should be able to reject the null hypothesis (a given explanation method is no better than random choice) with high confidence. ”
> In this section we evaluate the quality of IROF as an evaluation method. Intuitively, a good evaluator should be able to differentiate between a random baseline and an explanation method with high certainty. Motivated by this, we evaluate IROF with a two-sided t-test on a number of explanation methods.
> -          On page 6, forty images are considered in the text while fifty images are considered in the caption of Table 1. Please align the number with the correct one.
> Thank you for pointing this out. We fixed the table caption.
> [1] Ancona, Marco, et al. "Towards better understanding of gradient-based attribution methods for Deep Neural Networks." 6th International Conference on Learning Representations (ICLR 2018). 2018.

---

> > ### Comment · AnonReviewer1 · 2019-11-15
> > **Final score**
> >
> > After reading the response of the authors (thank you for them) I think I can increase my score.

---

### Official Review · AnonReviewer3 · 2019-10-21
**Official Blind Review #3**

**Rating:** 3

**Review:**

The paper has two main messages: 1- Averaging over the explanation (saliency map in the case of image data) of different methods results in a smaller error than an expected error of a single explanation method. 2- Introducing a new saliency map evaluation method by seeking to mitigate the effect of high spatial correlation in image data through grouping pixels into coherent segments. The paper then reports experimental results of the methods introduced in the first message being superior to existing saliency map methods using the second message (and an additional saliency map evaluation method in the literature). They also seek to magnify the capability of the 2nd message's evaluation method by showing its better capability at distinguishing between a random explanation and an explanation method with a signal in it.


I vote for rejecting this paper for two main reasons: the contributions are not enough for this veneue, and the paper's introduced methods are not backed by convincing motivations. The first message of the paper is trivial and cannot be considered as a novel contribution: the ''proof'' is basically the error of the mean is smaller than the mean of the errors. Additionally, this could have been useful if the case was that there was a need for playing the safe-card: that is, all of the existing methods have equal saliency map error and averaging will decrease the risk. Not only authors do not provide any evidence but also both the experimental results of the paper itself (results in Table 2 and Fig 4 are disproving this assumption) and the existing literature disprove it. Even considering this assumption to be correct, the contribution is minimal to the field and benefits of averaging saliency maps have been known since the SmoothGrad paper. The second contribution is an extension of existing evaluation methods (e.g. SDC) where instead of removing (replacing by mean) individual pixels, the first segment the image and remove the segments. The method, apart from being very similar to what is already there in the literature, is not introduced in a well-motivated manner. The authors claim that their evaluation method is able to circumvent the problem with removing individual pixels (which is the removed information of one pixel is mitigated by the spatial correlations in the image and therefore will not result in a proportional loss of prediction power) by removing ''features'' instead. Their definition of a feature, though, are segments generated by simple segmentation methods. There is a long line of literature showing the incorrectness of this assumption; i.e. a group of coherent nearby pixels does not necessarily constitute a feature seen by the network and does not necessarily remove the mentioned problem of the high correlation of pixels. This method does not remove "the interdependency of inputs" for the saliency evalatuion metric. Even assuming the correctness of this assumption, the contribution over what already exists in the literature is not enough for this venue.

A few suggestions:

* The authors talk about a ''true explanation''. This concept needs to be discussed more clearly and extensively. What does it mean to be a true evaluation? It is also important to prove that the introduced evaluation metric of IROF would assign perfect score for a given true explanation.

* The mentioned problem of pixel correlations that IROF seeks to mitigate is also existing in other modalities of data and the authors do not talk about how IROF could potentially be extended.

* The qualitative results in the text and the appendix do not show an advantage. It would be more crips if the authors could run simple tests on human subjects following the methods in the previous literature.

* There are many many grammatical and spelling errors in the paper. The font size for Figures is very small and unreadable unless by zooming in.

* Many of the introduced heuristics are not backed by evidence or arguments. One example is normalizing individual saliency maps between 0-1 which can naturally be harmful; e.g. aggregating a noisy low-variance method with almost equal importance everywhere (plus additive noise) and a high-variance one which does a good job at distinguishing important pixels - AGG-VAR will not mitigate this issue.

One question:

The authors introduced aggregation as a method for a ''better explanation''. It has been known that another problem with saliency maps is robustness: one can generate adversarial examples against saliency maps. It would be an interesting question to see whether aggregation would improve robustness rather than how good the map itself is.

**Experience Assessment:**

I have published in this field for several years.

**Review Assessment: Checking Correctness Of Derivations And Theory:**

I carefully checked the derivations and theory.

**Review Assessment: Checking Correctness Of Experiments:**

I carefully checked the experiments.

**Review Assessment: Thoroughness In Paper Reading:**

I read the paper thoroughly.

---

> ### Author Response · Authors · 2019-11-09
> **Author's response I**
>
> We thank the reviewer for their detailed review. We address their concerns below.
>
> “The first message of the paper is trivial and cannot be considered as a novel contribution: the ''proof'' is basically the error of the mean is smaller than the mean of the errors. Additionally, this could have been useful if the case was that there was a need for playing the safe-card: that is, all of the existing methods have equal saliency map error and averaging will decrease the risk. Not only authors do not provide any evidence but also both the experimental results of the paper itself (results in Table 2 and Fig 4 are disproving this assumption) and the existing literature disprove it.”
>
> As we clearly state and as noted by the reviewer,  the bias-variance decomposition in Eq. (1) shows that the error of the mean is less than or equal to the mean error over methods. We can not say anything about the error of the mean relative to the minimum error method. Please note, the result does not rely on all methods having equally large errors nor does it make assumptions about correlation structure of the explanations. We agree with the reviewer that this "proof" is well-known in other contexts. In particular we note, that Eq. (1) is an instance of standard bias-variance theory.  Yet, its relevance for explainability is new. We decided to give a detailed derivation in the Appendix A1, following comments from readers of earlier versions of the manuscript.
>
> We then go on to show, that in practice an aggregate also outperforms the best unaggregated method, i.e. has an error smaller than the minimum error.
> As another reviewer noted, an interesting idea for the future would be to study which minimum set of methods are useful to an aggregate. We agree with you that an interesting direction for that would be to give theoretical guarantees based on the similarity of the methods.
>
> “Even considering this assumption to be correct, the contribution is minimal to the field and benefits of averaging saliency maps have been known since the SmoothGrad paper. “
>
> Both, SmoothGrad and AGG-Mean aggregate explanations. At first glance, we agree that they do look similar. However, the two approaches differ in one vital aspect:
> SmoothGrad averages explanations from the same explanation method. This reduces the variance. AGG-Mean and AGG-Var averages over different explanation methods. This reduces the bias. As seen in our experiments, both AGGMean and AGGVar comfortably outperform SmoothGrad.
> Interestingly, SmoothGrad is outperformed by the unaggregated saliency map  on MNIST. On MNIST, AGGMean and AGGVar also do not outperform the best unaggregated method. It seems as if aggregation is more beneficial on high-dimensional datasets than on low-dimensional datasets.
>
>
> “The second contribution is an extension of existing evaluation methods (e.g. SDC) where instead of removing (replacing by mean) individual pixels, the first segment the image and remove the segments.”
>
> We cite several other approaches to evaluation in our paper. A the time [4], which introduced SDC,  had not been published yet (now at NeuRIPS 2019).
> As both other reviewers noted, we validate our results with extensive experiments, including on the validity of IROF as an evaluation method.
> The only other work that we are aware of that evaluates the evaluation method they introduced is [5].
>
> The reviewer asks what is meant by a ‘true explanation’ and a ‘true evaluation’. We agree that these are concepts mainly needed for theoretical analysis and for gaining intuition. There is ample need for clarifying the objectives and evaluation in this
> field. See additional comments below. We feel that our contribution makes an important step in this direction by proposing and evaluating a systematic and objective way to evaluate a new explanation method. Importantly, our work also does not have a large overhead involved, an aspect that becomes more and more important.
>
>
> “Their definition of a feature, though, are segments generated by simple segmentation methods. There is a long line of literature showing the incorrectness of this assumption; i.e. a group of coherent nearby pixels does not necessarily constitute a feature seen by the network and does not necessarily remove the mentioned problem of the high correlation of pixels.”
>
> We respectfully disagree with you about this. We are far from the only ones using segments as a rough approximation for features. The most recent in the context of neural networks is in fact the paper you mentioned on SDC  [4].

---

> > ### Author Response · Authors · 2019-11-09
> > **Author's response (continued)**
> >
> > “The authors talk about a ''true explanation''. This concept needs to be discussed more clearly and extensively. What does it mean to be a true evaluation? It is also important to prove that the introduced evaluation metric of IROF would assign perfect score for a given true explanation.”
> >
> > The existence of a true explanation is an assumption that is implicitly made in all papers about explainability. Roughly speaking, all explanation methods aim to rank input dimensions/features according to their importance for classification. IROF moves away from measuring importance in the (arbitrary) pixel representation and is a first step towards explanation evaluation based on objects instead.
> > The precise value of a perfect score depends on the dataset and the neural network being evaluated.
> >
> > We talk about the criterium of a good evaluation method in 4.2. We do not mention the concept of a ‘true evaluation’.
> >
> > “The qualitative results in the text and the appendix do not show an advantage. It would be more crips if the authors could run simple tests on human subjects following the methods in the previous literature. “
> >
> > We agree that we did not give great weight to human evaluation of explanations. This is by design. For one, the evaluation of neural network explanations hinges on the assumption that the human subject and the neural network rely on the same features for classification, i.e. that the explanations should align. As recent works show, this assumption may not hold up [2,3].
> >
> > Additionally, humans are notoriously easy to fool and are biased towards explanations that look crisp. See f.e. examples in [1]. At present we have not seen experimental designs that can leverage the complexity of working with human subjects, this is the topic of ongoing work.
> >
> > “Many of the introduced heuristics are not backed by evidence or arguments. One example is normalizing individual saliency maps between 0-1 which can naturally be harmful; e.g. aggregating a noisy low-variance method with almost equal importance everywhere (plus additive noise) and a high-variance one which does a good job at distinguishing important pixels - AGG-VAR will not mitigate this issue. “
> >
> > You are absolutely correct in that normalizing between 0-1 would give higher weight to low-variance methods, which is why we don’t do this.
> > As noted in section 3.1, we normalize all input heatmaps such that the sum of all positive relevance is one.
> > Additionally we clip all heatmaps at 0. As we note in section 4.1, this resulted only in negligible difference between the clipped and non-clipped version.
> >
> >
> > “There are many many grammatical and spelling errors in the paper.”
> >
> > We are sorry that we left the reviewer with the impression that the paper has  “many many grammatical and spelling errors”.  We will definitely prioritize this issue in the revisions.
> > Given the international aspect of our community, the majority of authors are not native English speakers. We hope the reviewer will let us know if there are mistakes that impede understanding.
> >
> > “The font size for Figures is very small and unreadable unless by zooming in.”
> > Thank you for pointing this out. We increased the font size in the revised manuscript.
> > We supply large version of the figures in the supplements. In the updated version of the papers we increased the font size for the figures.
> >
> > “The authors introduced aggregation as a method for a ''better explanation''. It has been known that another problem with saliency maps is robustness: one can generate adversarial examples against saliency maps. It would be an interesting question to see whether aggregation would improve robustness rather than how good the map itself is.”
> > Thank you for the insightful suggestion. We note that attacks on explanation methods typically concern specific explanation methods, thus there is some hope that the aggregation would be much more robust to attacks.
> >
> > [1] Adebayo, Julius, et al. "Sanity checks for saliency maps." Advances in Neural Information Processing Systems. 2018.
> > [2] Geirhos, Robert, et al. "ImageNet-trained CNNs are biased towards texture; increasing shape bias improves accuracy and robustness." (2018).
> > [3] Ilyas, Andrew, et al. "Adversarial Examples Are Not Bugs, They Are Features." arXiv preprint arXiv:1905.02175 (2019).
> > [4] Gohorbani, Amirata, et al. "Towards automatic concept-based explanations." (2019).
> > [5] Hooker, Sara, et al. "Evaluating feature importance estimates." arXiv preprint arXiv:1806.10758 (2018).

---

> > > ### Comment · AnonReviewer3 · 2019-11-15
> > > **Rebuttal**
> > >
> > > Thanks for your comprehensive response. Due to the provided clarifications, and having the mentioned novelty concerns in mind, I have changed my score accordingly.
> > >
> > > Just a small point:
> > >
> > > "Additionally, humans are notoriously easy to fool and are biased towards explanations that look crisp. See f.e. examples in [1]. At present we have not seen experimental designs that can leverage the complexity of working with human subjects, this is the topic of ongoing work."
> > >
> > > This does not sound like a valid reason for not conducting human-subject experiments. The goal of explainability research is to provide human-understandable interpretations.

---

### Official Review · AnonReviewer4 · 2019-10-28
**Official Blind Review #4**

**Rating:** 8

**Review:**

This paper, inspired by the established technique of model ensembling, proposes two methods (AGG-Mean and AGG-Var) for aggregating different model explanations into a single unified explanation. The authors mathematically prove that the derived explanation is guaranteed to be more truthful than the average performance of the constituent explanations. In practice, the aggregation consistently outperforms *all* individual explanations, not just their aggregated performance. Additionally, the paper introduces a new quantitative evaluation metric for explanations, free of human intervention: IROF (Incremental Removal of Features) incrementally grays out the segments deemed as relevant by an explanation method and observes how quickly the end-task performance is degraded (good explanations will cause fast degradation). Solid validation confirms that the IROF metric is sound.

I support paper acceptance. The experimental section is particularly strong, and makes a convincing argument for both the aggregation methods and the IROF metric. Even though I am not very familiar with the explainability literature and I would not be able to point out an omitted baseline for instance, the wide range of model architectures and aggregated explanation techniques makes a solid case. I appreciate the experiments on low-dimensional input, where the authors are deliberately showing a scenario in which their method does not score huge gains; this brings even more credibility to the paper. The presentation itself is clear, and there are no language or formatting issues.

The only obvious downside of AGG-Mean and AGG-Var is that one would have to implement and run all constituent evaluation methods, which is expensive. Just as an idea for future work: given N explanation methods, one could ablate away one method at a time, thus getting an idea of whether any of the N explanations are redundant in the presence of others. Recommending a minimal set of useful explanation methods to the NLP community would then decrease the overall complexity of replicating the end-to-end explanation system.

**Experience Assessment:**

I do not know much about this area.

**Review Assessment: Checking Correctness Of Derivations And Theory:**

I assessed the sensibility of the derivations and theory.

**Review Assessment: Checking Correctness Of Experiments:**

I assessed the sensibility of the experiments.

**Review Assessment: Thoroughness In Paper Reading:**

I read the paper at least twice and used my best judgement in assessing the paper.

---

> ### Author Response · Authors · 2019-11-09
> **Author's response**
>
> We thank the reviewer for their positive feedback and ideas for further work.
> “The only obvious downside of AGG-Mean and AGG-Var is that one would have to implement and run all constituent evaluation methods, which is expensive. “
> In regards to needing to implement and run all constituent explanation methods: The backpropagation-based methods we use for the aggregation are known for being comparatively fast to compute. To illustrate, obtaining an explanation with LIME, which is a sample-based explanation method, takes more time than obtaining an aggregation with all considered backpropagation -based methods.
> “Just as an idea for future work: given N explanation methods, one could ablate away one method at a time, thus getting an idea of whether any of the N explanations are redundant in the presence of others. Recommending a minimal set of useful explanation methods to the NLP community would then decrease the overall complexity of replicating the end-to-end explanation system.”
> We also agree with the reviewer that more complete ablation studies are interesting, complementing the mini-ablation study we have presented in Appendix A2,  where we evaluate combinations of two methods.

---

### Author Response · Authors · 2019-11-09
**Changes in the revision**

We would like to thank all reviewers for their time and effort. We have responded to their concerns below, and made the following changes to the manuscript as a result:
increased font size for figure 2 and 4
moved second row of figure 2 to supplements due to space
changed caption for figure 1
corrected number in table 1
corrected typos
changed first paragraph of section 4.2

---

### Decision · Program_Chairs · 2019-12-19

**Decision:**

Reject

**Comment:**

This paper describes a new method for explaining the predictions of a CNN on a particular image. The method is based on aggregating the explanations of several methods.   They also describe a new method of evaluating explanation methods which avoids manual evaluation of the explanations.

However, the most critical reviewer questions the contribution of the proposed method, which is simple.  Simple isn't always a bad thing, but I think here the reviewer has a point.  The new method for evaluating explanation methods is interesting, but the sample images given are also very simple -- how does the method work when the image is cluttered?   How about when the prediction is uncertain or wrong?